# Untargeted Lipidomics and Chemometric Tools for the Characterization and Discrimination of Irradiated Camembert Cheese Analyzed by UHPLC-Q-Orbitrap-MS

**DOI:** 10.3390/foods12112198

**Published:** 2023-05-30

**Authors:** Michele Tomaiuolo, Valeria Nardelli, Annalisa Mentana, Maria Campaniello, Rosalia Zianni, Marco Iammarino

**Affiliations:** Laboratorio Nazionale di Riferimento per il Trattamento degli Alimenti e dei loro Ingredienti con Radiazioni Ionizzanti, Istituto Zooprofilattico Sperimentale della Puglia e della Basilicata, Via Manfredonia, 20-71121 Foggia, Italy; michele.tomaiuolo@izspb.it (M.T.); valeria.nardelli@izspb.it (V.N.); maria.campaniello@izspb.it (M.C.); rosalia.zianni@izspb.it (R.Z.); marco.iammarino@izspb.it (M.I.)

**Keywords:** lipidomics, food irradiation, UHPLC-Q-Orbitrap-MS, chemometric analysis, Camembert cheese

## Abstract

In this work, an investigation using UHPLC-Q-Orbitrap-MS and multivariate statistics was conducted to obtain the lipid fingerprint of Camembert cheese and to explore its correlated variation with respect to X-ray irradiation treatment. A total of 479 lipids, categorized into 16 different lipid subclasses, were measured. Furthermore, the identification of oxidized lipids was carried out to better understand the possible phenomena of lipid oxidation related to this technological process. The results confirm that the lipidomic approach adopted is effective in implementing the knowledge of the effects of X-ray irradiation on food and evaluating its safety aspects. Furthermore, Partial Least Squares-Discriminant Analysis (PLS-DA) and Linear Discriminant Analysis (LDA) were applied showing high discriminating ability with excellent values of accuracy, specificity and sensitivity. Through the PLS-DA and LDA models, it was possible to select 40 and 24 lipids, respectively, including 3 ceramides (Cer), 1 hexosyl ceramide (HexCer), 1 lysophosphatidylcholine (LPC), 1 lysophosphatidylethanolamine (LPE), 3 phosphatidic acids (PA), 4 phosphatidylcholines (PC), 10 phosphatidylethanolamines (PE), 5 phosphatidylinositols (PI), 2 phosphatidylserines (PS), 3 diacylglycerols (DG) and 9 oxidized triacylglycerols (OxTG) as potential markers of treatment useful in food safety control plans.

## 1. Introduction

Camembert is a surface mould-ripened cheese of French origin characterized by white or light-grey rind, roughly 3 mm thick [1], obtained through the activity of *Penicillium camemberti* (or *P. candidum*), sprayed on the cheese surface or directly inoculated into the milk during cheese manufacture. The presence of this mould brings unique aromas and distinctive sensory characteristics to Camembert cheese [2].

Many physical, microbiological and biochemical changes occur during the ripening of Camembert-type cheeses, and these modifications continue during their packaging and storage at 4–6 °C [2]. More specifically, the structural changes of these cheeses are related to protein matrix swelling, due to the centre-to-surface migration of minerals [3] while the microbiological and biochemical modifications initially concern phenomena of glycolysis, proteolysis and lipolysis and subsequently involve the metabolism of amino acids and fatty acids that determine the peculiar organoleptic characteristics of these foods [1,4].

Regarding the safety aspects, according to the EU regulation with a Community Directive [5], Camembert cheese is included in the ready-to-eat (RTE) food category, with a potential growth risk of *Listeria monocytogenes* [6]. This pathogen can also adhere to food processing surfaces forming biofilms, so its occurrence in the processing environment together with flawed hygiene practices could cause post-processing contamination of Camembert [7]. Therefore, it is reasonable to believe that the application of technological sanitization processes after packaging could guarantee the safety of these types of products. Besides the safety aspects, as well as other soft cheeses with surface mould, Camembert has a relatively short shelf-life, which depends on the production process, packaging, storage and distribution conditions. To the best of our knowledge, few solutions have been proposed to control microbial proliferation, preserving the nutritional components and sensory characteristics of Camembert cheese, such as the use of pasteurized milk and different combinations of ingredients (starter cultures and moulds) or methods of mould inoculation [8].

In this context, among non-thermal technologies, food X-ray irradiation represents a clean and safe valid alternative to preserve the hygienic quality of food and to extend the shelf-life of several foodstuffs [9], including dairy products [10,11], and in this regard, to date, it has been shown that Camembert cheeses manufactured from raw milk can be treated, at a maximum dose of 2.5 kGy [12] to reduce pathogens such as *L. monocytogenes* and *Salmonella spp.* [13]. However, in addition to microbial growth evaluation, the study of the potential for radiation-induced alteration of irradiated foods is of great importance for their acceptance on the market. In this context, the lipidomic approach has never been used in the characterization and discrimination of irradiated dairy products.

Lipids are a heterogeneous group of compounds involved in many biological functions as intermediates or products in signalling pathways, structural components of cell membranes and energy storage sources, and lipidomics is an extensive and comprehensive approach to the study of these compounds in biological systems, useful for many purposes, such as assessing the authenticity and adulteration of foods [14]. More specifically, untargeted lipidomic strategies focus on the analysis of all detectable lipids in a sample, in contrast with the targeted approach, which is the measurement of defined groups of lipids. The evaluation of the effects on the global lipidome of food when treated with technological processes, such as irradiation, can be accomplished only by using untargeted methods [15]. Moreover, this approach is the most powerful tool for the identification of new biomarkers and lipid mediators due to the possibility of identifying unknown but relevant lipids [16,17]. However, the understanding of untargeted *omics* is complicated due to the large amount of mass spectrometry data together with the complexity of data processing and interpretation, so both univariate and multivariate tests are being employed [18]. In combination with multivariate tests, classification models can be rendered to further isolate the most discriminative lipid species based on their relevance, i.e., sensitivity and specificity as predictive and treatment markers. Furthermore, it is very important to underline that a robust validation approach is required for these models [19].

In this work, the lipid composition of commercial Camembert cheese under irradiation treatment was evaluated via an untargeted lipidomic approach by means of Ultra-High-Performance Liquid Chromatography coupled with Quadrupole Orbitrap Mass Spectrometry (UHPLC-Q-Orbitrap-MS). A dose of 3 kGy, slightly higher than the legal limit, was chosen because, low doses, less and equal to this value, are recommended for the treatment of soft cheeses [20]. Particular emphasis was placed on chemometric analysis, involving supervised and unsupervised methods and subsequent validations of different models for discriminant analysis. The individuation of potential treatment markers is useful in food safety control plans.

## 2. Materials and Methods

### 2.1. Chemicals and Working Standard Solutions

Ammonium formate (NH_4_HCO_2_), isopropanol (IPA), water (H_2_O), acetonitrile (ACN) and formic acid (HCO_2_H) LC/MS grade were acquired from Carlo Erba Reagents (Cornaredo, MI, Italy). Chloroform (CHCl_3_) HPLC grade was purchased from Merck Life Science S.r.l. (Darmstadt, Germany), and methanol (MeOH) LC/MS grade was supplied by EMD Chemicals (Gibbstown, NJ, USA). 1,2,3-tripelargonoyl-glycerol (trinonanoin, 9:0-9:0-9:0-TAG) and the deuterated lipid internal standards, Equisplash™ Lipidomix^®^ 100 mg L^−1^, were purchased from Merck Life Science S.r.l. (Darmstadt, Germany). For analyses, a stock standard solution and a working standard solution of trinonanoin, 10,000 mg L^−1^ in CHCl_3_/MeOH (1:1, *v*/*v*) and 1000 mg L^−1^ in MeOH/CHCl_3_ (4:1, *v*/*v*), respectively, were used.

### 2.2. X-ray Irradiation Treatment

X-ray irradiation of cheeses was performed in the National Reference Laboratory of Istituto Zooprofilattico Sperimentale della Puglia e della Basilicata. The samples were placed into 500 mL carbon fibre tubes with a diameter of 80 mm. Irradiation was carried out in a room with an ambient temperature of 20 °C using a low-energy X-ray irradiator (RS-2400, Radsource Inc., Suwanee, GA, USA) operating at 150 kV and 45 mA. The average dose absorbed by the samples under X-ray irradiation was estimated with an alanine/electron paramagnetic resonance dosimetry system. A calibrated ionization chamber (Radcal Inc., Monrovia, CA, USA) was used to obtain the alanine signal dose amplitude calibration curve, and the uncertainty of the value of delivered dose was around 5%. For this investigation, a dose level of 3.0 kGy at a dose rate of approximately 2 kGy h^−1^ was used.

### 2.3. Sample Extraction

Four Camembert cheese samples of 250 g, produced from pasteurized milk and packaged in thin wooden boxes, were purchased in a local market and stored at 4 °C. Each cheese was divided into two portions: the first represented the control non-irradiated (CAM_NI), and the second was irradiated at 3 kGy (CAM_IRR), making a total of 8 samples, analyzed in triplicate. Lipid extraction was performed based on the Folch method [21], opportunely adapted to our matrices. Specifically, 300 µL of trinonanoin and 19 mL of CHCl_3_/MeOH solution (2:1, *v*/*v*) were added to a 1.0 g of sample, and the mixture was then vortexed with a TX4 Digital Vortex Mixer (Velp Scientifica, Usmate, Italy) at 600 rpm for 15 min and centrifuged using a BKC-DL5M centrifuge (Biobase Meihua Trading Co., Ltd., Jinan, China) at 1500 rpm for 30 min at 4 °C. Then, another 19 mL of CHCl_3_/MeOH solution (2:1, *v*/*v*) was added and the mixture was vortexed and centrifuged for 2 and 15 min, respectively. After that, 9.5 mL of H_2_O was added, and the mixture was kept overnight at 4 °C. Afterwards, the tube was centrifuged at 1500 rpm for 10 min at 4 °C, and then the lower phase with CHCl_3_ was filtered and the solvent was evaporated at 40 °C. Fifty mg of dry extract was dissolved in 5 mL of MeOH/CHCl_3_ (1:1, *v*/*v*) and, for injection in UHPLC-Q-Orbitrap-MS, the solution was then 5-fold diluted with MeOH/CHCl_3_ (4:1, *v*/*v*).

### 2.4. Untargeted Analysis

All analyses were performed using an Ultimate 3000 UHPLC coupled with a Q-Exactive Focus Orbitrap Mass Spectrometer (Thermo Fisher Scientific, Waltham, MA, USA) equipped with a heated electro-spray ionization (HESI) source. The chromatographic conditions and the analytical parameters are shown in Table 1.

In this study, a procedural blank, defined as Quality Assurance (QA), was used to assure the performance and final outcomes of the experiments [22]. QA was also useful in the *search* step and was inserted in the *alignment* dataset for Lipidsearch^TM^ elaboration. Quality control (QC), containing Equisplash™ Lipidomix^®^ and trinonanoin, was helpful to verify the system stability and the repeatability of the acquisitions, and it was analyzed every 10 injections. [23]. Finally, for conditioning the chromatographic system, a Pooled Sample (PS), prepared by pooling equal 150 µL aliquots of six lipid extracts, was ejected at the beginning of the analytical batch.

UHPLC-Q-Orbitrap-MS data were processed by LipidSearch^TM^ v4.2.2.7 software (Thermo Fisher Scientific, Waltham, MA, USA) based on accurate precursor ion mass and fragmentation features [24]. Detailed software parameters are reported in Table 1. Special attention was given to the identification of oxidized lipids by inserting “Oxid. GPL” in the database of the Lipidsearch^TM^ and by manual evaluation of MS/MS spectra using FreeStyle^TM^ v1.6 software (Thermo Fisher Scientific, Waltham, MA, USA).

### 2.5. Statistical Analysis

All statistical and chemometric analyses were performed thanks to the free software R v4.1.1 (R Development Core Team, Vienna, Austria, 2020), using in-house routines, partly based on the mdatools package [25].

#### Diagnostic Statistics

To quantify the ability of discriminant analysis, the following diagnostic parameters were used: *Q^2^*, *DQ*^2^, Sensitivity, Specificity, Accuracy and *AUROC*. Specifically, *Q^2^* estimates the fraction of the deviance explained by the model compared to the total deviance, and it is defined as one minus the ratio of the prediction error sum of squares (*PRESS*) over the total sum of squares (*TSS*) of the reference value y:$$Q^{2}=1-\frac{\sum_{i}{{(y}_{i}-{\hat{y}}_{i})}^{2}}{\sum_{i}{{(y}_{i}-{\overset{-}{y}}_{i})}^{2}}=1-\frac{PRESS}{TSS}$$

When applied to the discriminant analysis, the previous equation can cause the value of *PRESS* to increase in an unjustified manner, and therefore decrease the estimate of *Q^2^*. When the value predicted by the model is close to the discrimination limit, it is right for *PRESS* to increase. However, when, for example, the discrimination level is 0, the reference value *y* is 1 and the predicted value $\hat{y}$ is 1.3, the perfect discrimination is obtained and the residual $y-\hat{y}$ would contribute improperly to the calculation of *PRESS*. An alternative parameter that takes this phenomenon into account calculates the value of *PRESS* in the following way [26]:$${PRESSD}_{class=-1}=\sum_{{\hat{y}}_{i}<-1}{\left(y_{i}-{\hat{y}}_{i}\right)}^{2}$$
$${PRESSD}_{class=+1}=\sum_{{\hat{y}}_{i}<-1}{\left(y_{i}-{\hat{y}}_{i}\right)}^{2}$$
$$PRESSD={PRESSD}_{class=+1}+{PRESSD}_{class=-1}$$
$${DQ}^{2}=1-\frac{PRESSD}{TSS}$$

Moreover, sensitivity refers to the fraction of CAM_IRR that have been classified as irradiated, specificity refers to the fraction of CAM_NI that have been classified as non-irradiated and accuracy refers to the fraction of correctly classified samples, as follows:$$Sensitivity=\frac{true\;CAM\_IRR}{true\;CAM\_IRR+false\;CAM\_NI}$$
(1)$$Specificity=\frac{true\;CAM\_NI}{true\;CAM\_NI+false\;CAM\_IRR}$$
(2)$$Accuracy=\frac{true\;CAM\_IRR+true\;CAM\_NI}{total\;samples}$$

Finally, the Receiver Operator Characteristic (ROC) parameter combines sensitivity and specificity. In particular, the ROC curve reports sensitivity on the ordinate and 1-specificity on the abscissa at different thresholds [27]. This curve was estimated by *AUROC*, which is the calculation of the area under the ROC curve.

## 3. Results and Discussion

### 3.1. Lipid Identification and Characterization

Sixteen lipid subclasses, including eventually related oxidized forms, were extracted from Camembert cheese by the Folch procedure and then identified by UHPLC-Q-Orbitrap-MS analysis. Specifically, in positive ion mode, 345 triacylglycerols (TG) and 42 related oxidized forms (OxTG) were identified as +NH_4_ or +Na and +NH_4_ or +H adducts, respectively, and distinguished by the composition of fatty acids and positional isomers. Moreover, nine diacylglycerols (DG) as +Na adducts and one related oxidized form (OxDG) as +H adduct, one bismethyl phosphatidic acid (BisMePA) as +Na adduct, and one phosphatidylethanol (PEt) as +H adduct and cholesterol ester (ChE) as +H−H_2_O adduct, were measured. In negative ion mode, 13 ceramides (Cer) and 2 related oxidized form (OxCer), 7 hexosyl ceramides (4 Hex1Cer and 3 Hex2Cer) and 1 related oxidized form (OxHex1Cer), 1 monogalactosyldiacylglycerol (MGDG), 1 lysophosphatidylcholine (LPC), 11 phosphatidylcholines (PC) and 7 sphingomyelins (SM), all as +HCOO adducts together with 1 lysophosphatidylethanolamine (LPE), 5 phosphatidic acids (PA), 15 phosphatidylethanolamines (PE), 9 phosphatidylinositols (PI) and 6 phosphatidylserines (PS) as −H adducts, were identified. Full details on the individual lipids identified are listed in the supplementary materials (Folder_01 of Mendeley Data [28]) associated with this manuscript. Figure 1 displays the qualitative fingerprint (in number and type) of CAM_NI, which did not change after irradiation, so there was no variation in lipid components between CAM_IRR and CAM_NI. On the other hand, differences in the abundance of specific lipids were observed and considered for chemometric analysis.

#### Oxidized Lipids

The identification of oxidized lipids is useful for understanding the effects of a technological treatment, such as X-ray irradiation, which involves the formation of hydroxyl radicals generated by the homolysis of water [29]. These radicals are able to abstract an allylic hydrogen atom in lipids containing two or more double bonds. Successively to the addition of O_2_, corresponding to the initiation phase of lipid peroxidation, the propagation phase occurs by lipid-lipid interactions resulting in a magnification of radical formation [30]. During this propagation phase, unsaturated lipids are oxidized into the corresponding alkoxy and peroxy radicals. These radicals are further degraded into secondary compounds, including alcohols, ketones, epoxides, aldehydes and hydrocarbons, and their formation is responsible for sensory alterations associated with lipid oxidation, such as odours and flavours [31,32].

In our paper, the identification of oxidized lipids was performed with the support of LipidSearch^TM^ v4.2.2.7 software that contains an inbuilt “Oxid. GPL” database, covering oxidative modifications of phospholipids, triacylglycerols, diacylglycerols and fatty acids. This software is capable of identifying the simple addition of oxygen on the alkyl chain or the fragmentation of fatty acid chains with the formation of corresponding aldehyde-, carboxyl- and methyl ester groups [33]. These molecules are identified by a specific annotation: “+O” indicating the addition of OH to fatty acids; “+OX” indicating the presence of an epoxide group; “CHO”, “COOH” or “COOCH3” indicating the addition of a terminal aldehyde, carboxylic acid or methyl ester to fatty acids, respectively. Using the same *search* and *alignment* parameters described in the data processing section, 16 “+O” or “+OX” long-chain OxTG, with a number of carbon atoms equal to or greater than 18, were identified. Five of these were also detected in their non-oxidized form. A list of 26 short-chain OxTG including 16 CHO, 8 COOH and 2 COOCH3, generated by cleavage of long-chain unsaturated fatty acids, was identified. All OxTG were grade “A” or “B”.

In addition, one OxDG (DG(38:5+OO)) of grade “C” in positive ionization mode, two OxCer (Cer(t17:0_25:0+O)) and Cer(t18:0_25:0+O)) and one OxHex1Cer (Hex1Cer(m35:3+2O)), respectively, of grade “B” and “C” in negative ion mode, were identified. Note that OxCer grade “B” is linked to the identification of the neutral loss, NL [H_2_O, Amide (25:0+O)] (CalcMz 267.23295), and the oxidized fragment of the 25:0 fatty acid chain. Long and short-chain OxTG, OxDG, OxCer and OxHex1Cer were measured in both CAM_NI and CAM_IRR, suggesting that previous technological treatments, such as pasteurization and/or enzymatic processes, can be the source of these molecules, while the X-ray dose employed in this study was not sufficient to generate specific oxidized lipids as new markers of irradiation treatment.

Nevertheless, the results showed differences between CAM_IRR and CAM_NI in the amount of these oxidized lipids. However, oxidative lipidomics is an emerging discipline without guidelines for proper annotation, and therefore only the most confident identifications, with the support of manual evaluation of MS/MS spectra, were retained.

### 3.2. Chemometrics

#### 3.2.1. Data Exploration

In order to verify the presence of both possible first aggregations and outliers, an unsupervised PCA was performed on the dataset from negative and positive ion modes (Figure 2A,C). The percentages of variance explained by the PC1 and PC2 for the two datasets were similar: 36.6% and 29.9% for the negative dataset and 36.5% and 28.4% for the positive dataset. However, in these PCA, no distinct groupings of CAM_IRR and CAM_NI were highlighted. These results suggested that an unsupervised approach by evaluating only two principal components was not suitable for discrimination between CAM_IRR and CAM_NI. Moreover, normalized orthogonal distance and normalized Hotelling T^2^ distance were used, considering a significance level of 0.01, to identify the possible outliers (Folder_02 of Mendeley Data [28]) and, for both datasets, no samples beyond the critical limit were found; therefore, all experimental data were used for the elaboration of discriminant models.

#### 3.2.2. PLS-DA Elaboration

##### Data Pre-Processing

Partial least squares discriminant analysis (PLS-DA) is the most used classification method in metabolomics [34]. Firstly, the selection of the lipids as significant variables to use for the PLS-DA model was carried out through both the volcano plot and the VIP score. In the volcano plot, many lipids were clustered in a cloud below a threshold value in both the negative and positive datasets (Figure 3). These molecules did not produce significant differences in the ANOVA test (*p* > 0.05) between CAM_IRR and CAM_NI, with a deviation of their mean values close to zero.

As for the VIP score, this value is significant for evaluating the contribution of a given variable to the whole model, as the higher the VIP value the more important the contribution for the classification [35]. Consequently, the lipids having a *p*-value ≤ 0.05 in ANOVA, corresponding to a threshold value of −log10*p*-value ≥ 1.3 in the volcano plot, and a cut-off value of 1 for VIP score [36] were considered to be potentially significant for the separation of CAM_IRR and CAM_NI (Figure 4). In this way, 40 lipids were selected as important contributors, including 8 OxTG, 2 DG, 3 Cer, 1 Hex1Cer, 1 LPC, 1 LPE, 3 PA, 4 PC, 10 PE, 5 PI and 2 PS (Table 2). The results showed a decrease in short-chain OxTG containing aldehyde, carboxylic acid or methyl ester groups after irradiation, so the involvement of these molecules in further oxidation steps can be hypothesized. On the other hand, a long-chain OxTG, TG (18:2+O_18:0_18:0), increased in CAM_IRR. Note that this trend occurred for all identified long-chain OxTG, suggesting that they were produced by lipid oxidation phenomena due to irradiation treatment. As for the other selected molecules, DG and polar lipids (i.e., phospholipids) decreased with irradiation, while the subclass of Cer, which is another minor lipid class of dairy products, which is generally considered structurally similar to sphingolipids and glycolipids [37], slightly increased. Detailed information on potential markers, including the mass of the compounds, error, molecular formula and their fragmentation pattern, is listed in the Folder_01 of Mendeley Data [28].

##### PLS-DA in Double Cross-Validation

PLS-DA can be applied to datasets with a number of predictors (lipids) higher than the number of objects (runs), as often occurs in metabolomics studies, and therefore is not affected by the predictor collinearity problem. In the PLS-DA model setup, there are two critical points: the selection of the optimal number of latent variables (#LV) and the assessment of the overall quality of the model [34]. In our data elaboration, a double cross-validation algorithm was used involving the split of the dataset into two nested loops, the inner loop (CV1) and outer loop (CV2), with the aim of optimizing the model and defining the diagnostic statistics [38]. In general, the PLS-DA model improves when *Q^2^*, *DQ^2^*, accuracy, sensitivity, specificity and *AUROC* increase, while for the root mean squared error of cross-validation (*RMSECV*), which indicates how closely a model predicts the measured values, the optimal targeted value is the lowest. Considering also that diagnostic statistic parameters, when taken individually, can determine a diverse number of optimal #LV in the same model, another index was formulated, called Efficiency index (*I_eff_*) (Equations (1) and (2)), which was defined as the sum of diagnostic statistics calculated in CV1, i.e., *Q^2^*, *DQ^2^*, accuracy, sensitivity, specificity, *AUROC* and a transformation of *RMSECV*
${(t}_{RMSECV})$ [20].
(3)$$I_{eff}=Q^{2}+{DQ}^{2}+accuracy+sensitivity+specificity+AUROC+t_{RMSECV}$$
where:(4)$$t_{RMSECV}=\frac{{RMSECV}_{max}-RMSECV}{{RMSECV}_{max}-{RMSECV}_{min}}\cdot \frac{{RMSECV}_{min}}{RMSECV}$$

*I_eff_* overcomes the subjectivity of choice of the optimal #LV number, setting it at the maximum value of this index (Folder_03 of Mendeley Data [28]). The entire double cross-validation process was repeated 200 times to calculate the average performance value of the model and to estimate its robustness (Table 3). The results obtained highlighted the strong discriminating ability of the PLS-DA with slightly better results for data obtained in negative ion mode. Both CAM_IRR and CAM_NI were correctly classified with an accuracy value of 99.9% and 98.8% for negative and positive datasets, respectively. The *AUROC* also showed a value close to 1, confirming the good separation of the distribution of the predicted values for the two groups of data. Finally, the dispersion of the diagnostic statistics in the 200 repetitions of the double cross-validation was greater in the data measured in positive ion mode (Folder_03 of Mendeley Data [28]). The score plot in Figure 2B,D highlights the discriminating ability of the PLS-DA model, showing CAM_IRR and CAM_NI in two clearly separated clusters.

In this study, predicted uncertainties were also estimated by means of bootstrap, stratified random subsampling, Kennard–Stone sampling and permutation test using the optimal number of #LV at eight for the negative dataset and five for the positive dataset. These values were obtained by double cross-validation (Table 3).

##### Bootstrap

The bootstrap algorithm is a resampling technique in which the user decides the number of iterations. In each iteration, for a dataset of *n* objects, *n* samples are chosen for training with replacement. The validation subset is formed by the rest of the samples [39]. The number of iterations performed was 10,000, obtaining good values of e of sensitivity, specificity and accuracy (Table 3).

##### Stratified Random Subsampling

Both the number of iterations and the percentage of training and validation subsets can be decided by using the stratified random subsampling validation method. Training samples are randomly chosen, and the rest are used for validation, without resampling. Samples could be found many times in the validation subset [40]. The proportion between the number of CAM_IRR and CAM_NI samples was preserved in both the validation and training sets and the dataset was split into training and validation subsets using a 3:1 ratio. The procedure was repeated 10,000 times obtaining very good sensitivity, specificity and accuracy (Table 3).

##### Kennard–Stone Sampling

The Kennard–Stone algorithm selects samples with large Euclidean distances between them [41]. Sampling was carried out both with the stratified method, preserving the proportion between the number of CAM_IRR and CAM_NI samples, and without the stratified method. In both methods, all samples of the validation set were correctly classified (Table 3).

#### 3.2.3. LDA Elaboration

The Linear Discriminant Analysis model was used as an alternative supervised technique to discriminate between CAM_IRR and CAM_NI. Considering that, in the LDA algorithm application, for each category, the number of variables must be no greater than the objects, a selection of the molecules was made on the basis of the results of the volcano plot (Figure 3). A total of 24 lipids were selected to be potentially discriminant, as listed in Table 2. These discriminatory lipids were composed of nine OxTG, three DG, two PC, five PE, three PI and two PS (Folder_01 of Mendeley Data [28]). With respect to the molecules selected for double cross-validation, one OxTG, TG (18:1+O_18:1_18:1) and one DG (8:0_14:0) were included, since they showed the same trend previously described. The Lilliefors normality test was conducted with a *p*-value of 0.05 on the area values of the single molecules, for CAM_IRR and CAM_NI separately, and a log transformation was applied to the molecules that failed the test to improve the behaviour of the variable to a normal distribution. The model performances were estimated by sensitivity, specificity and accuracy values of a cross-validation process repeated 10,000 times. The construction of the training set and validation set were obtained with a stratified sampling method to preserve the proportion between the two classes of samples (irradiated and not irradiated)

Similar to PLS-DA, this LDA model showed good discriminating ability between samples before and after X-ray irradiation, with average values of sensitivity, specificity and accuracy of 93.7%, 97.9% and 95.8%, respectively, for the positive dataset, and 100% 93.0% and 96.3%, for the negative dataset.

#### 3.2.4. Permutation Test

The permutation test allows us to verify if the results obtained in the validation of the classification models depend on the simple case. In this test, the response variable is replaced with a permutation of it in order to obtain a random association between the response and predictors [38]. The permuted dataset (H0) distribution was estimated by generating 30,000 permutations of the original classification. For each of these 30,000 models, the number of misclassified samples (NMC_P_) was calculated using simple cross-validation. The average number of misclassified samples obtained in the double cross-validation phase for the PLS-DA model and cross-validation for the LDA model (NMC) was compared with the H0 distribution, and the *p*-value was calculated as:(5)$$p-value=\frac{1+\#({NMC}_{P}\leq NMC)}{N}$$
where #NMC_P_ ≤ NMC is the number of permuted models that generated a number of misclassified samples less than or equal to the average number of those in the validation phase. In PLS-DA, for both positive and negative datasets, a *p*-value of 6.6 × 10^−5^ was obtained, confirming that the discriminating ability of the model was not determined by random phenomena. The mean values of the H0 distribution were 12.5 (24 samples) and 13.4 (26 samples) for the positive and negative datasets, respectively, and they were compatible with the mean value of a binomial distribution with probability π = 0.5. A similar result was obtained with the LDA algorithm with a *p*-value of 6.6 × 10^−5^ for negative data and 1.0 × 10^−4^ for positive dataset (Folder_04 of Mendeley Data [28]).

## 4. Conclusions

In this study, a comprehensive lipidomic approach by means of UHPLC-Q-Orbitrap-MS and multivariate statistics was used to obtain the lipid fingerprint of Camembert cheese and to investigate how it varies when irradiated at 3 kGy. The results provided the lipid profile of this cheese, characterized by 479 lipids classified in 16 different lipid subclasses. Special attention was given to oxidized lipids, and the results demonstrated that the X-ray dose employed in this investigation did not lead to the formation of specific oxidized lipids or, in general, of new lipid molecules linked to this treatment. However, lipidomic analysis combined with chemometric modelling enabled the discrimination of irradiated versus non-irradiated samples and the selection of 42 lipids as potential treatment markers. Among the models tested, PLS-DA in double cross-validation showed the best discriminating ability. In conclusion, the results confirm the effectiveness of this *omic* approach for deepening knowledge of the effects of technological treatment on food, which is also helpful in food safety control plans. Further investigations on a larger sample size are needed to confirm the potential lipid markers identified.

## Figures and Tables

**Figure 1 foods-12-02198-f001:**
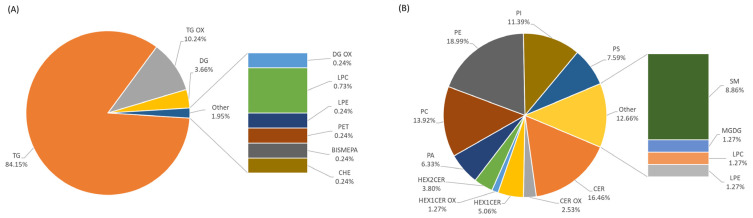
Qualitative lipid fingerprint of non-irradiated Camembert with the percentages related to the contribution of each lipid subclass. Lipids were identified in positive (**A**) and negative (**B**) acquisition modes.

**Figure 2 foods-12-02198-f002:**
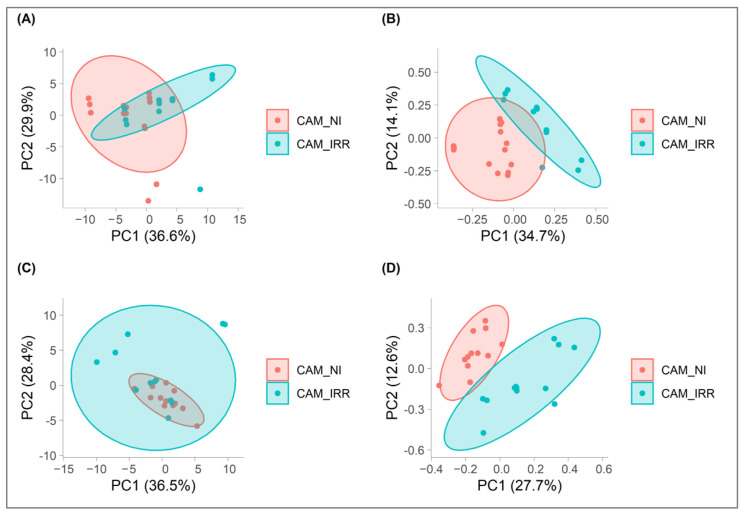
Score plots of PCA in negative (**A**) and positive (**C**) ion modes and PLS-DA in negative (**B**) and positive (**D**) ion modes of CAM_IRR and CAM_NI.

**Figure 3 foods-12-02198-f003:**
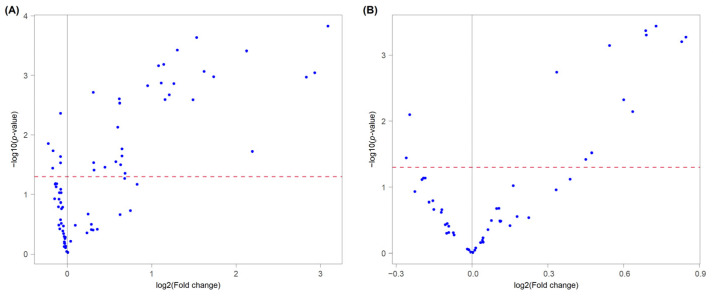
Volcano plot showing the most significant lipids by univariate analysis, identified in negative (**A**) and positive (**B**) ion modes.

**Figure 4 foods-12-02198-f004:**
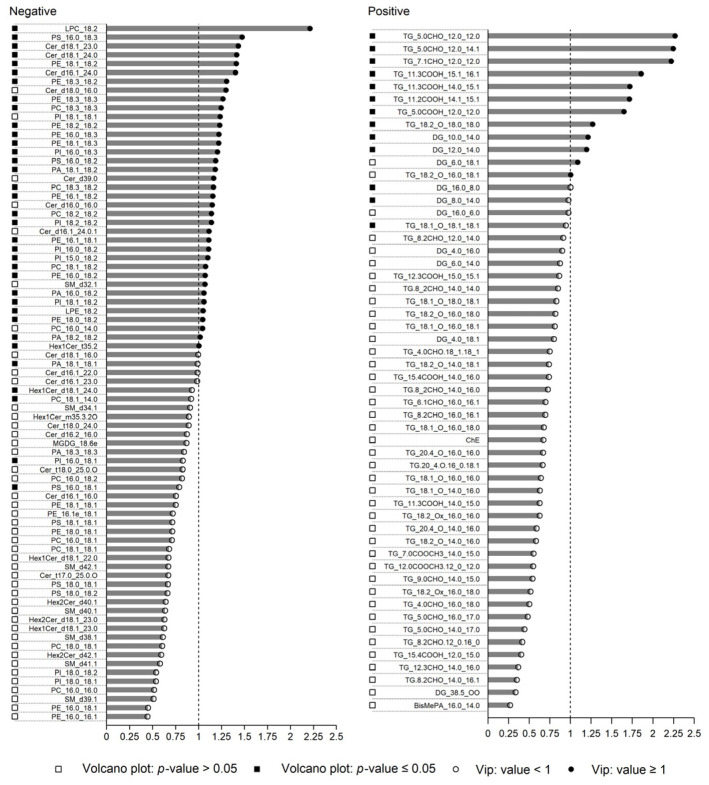
List of lipids identified in positive and negative ion mode with the corresponding *p*-value obtained from the volcano plot together with the VIP score to evaluate their significance for CAM_IRR discrimination. For better graphical visualization, TG are not shown as they are not significant for chemometric analysis.

**Table 1 foods-12-02198-t001:** Operating chromatographic conditions and MS setting of UHPLC-Q-Orbitrap-MS system and Lipidsearch^TM^ parameters.

Operating Chromatographic Conditions		MS Setting		Data Processing Lipidsearch^TM^ Software	
Sample temperature	18 °C	Scan range (*m*/*z*)	150–2000	*Search* parameters	
Column and security guard column	Accucore C30 column (150 × 2.1 mm 2.6 µm column, Thermo) with a security guard column ULTRA Cartridges UHPLC wide-pore C18 (AJ0-8769, 2 × 4.6 mm ID, with sub-2 m particles, Phenomenex)	Full scan resolution (FWHM)	70,000	Search Class	ALL Lipids
				Ions	+H; +NH4; +Na; +(CH3CH2)3NH; +(CH3)2NH2; +H−H2O; +H−2H2O; +2H (+) −H; +HCOO; +CH3COO; −2H; −CH3 (−)
Inject Volume	2 µL (+); 4 µL (−)	Multiple data-dependent (dd-MS^2^) scan resolution (FWHM)	17,500	Identification	Precursor tolerance: 5.0 (+); 8.0 (−) ppm
					Product tolerance: 8.0 (+); 10.0 (−) ppm
					m-Score threshold: 5.0
					Database: General; HCD; Oxid. GPL; labelled GPL, GL, SP, ChE
Phase A	ACN/H_2_O (60:40, *v*/*v*), 10 mM NH_4_HCO_2_ and 0.1% HCO_2_H			Search Filters	Top rank filter
					Main node filter: all isomer peak
					FA priority
					ID quality filter: A, B, C D
Phase B	IPA/ACN (90:10, *v*/*v*), 10 mM NH_4_HCO_2_ and 0.1% HCO_2_H	Spray voltage (kV)	3.4 (+); 3.3 (−)	*Alignment* Parameters	
Flow rate	270 µL min^−1^	Capillary temperature (°C)	290	Search Type	Product
Elution gradient		Auxiliary gas heater (°C)	290	Exp Type	LC-MS
Time (min)	Percentage of B (%)	Sheath gas (Arb)	32	Normalize type	None
0	25	Auxiliary gas (Arb)	8	Alignment method	Mean
4.0	43	Sweep gas (Arb)	0	R.T. Tolerance	0.1
4.1	55	S-lens RF level	50	Calculate unassigned peak area	On
12.0	65	AGC Target	1e6	Top rank filter	On
18.0	85	Stepped normalized collision energy	20, 30 (+); 25, 40 (−)	Main node filter	Main isomer peak
20.0	100	Maximum Injection time (ms)	50	m-score Threshold	5.0
26.0	100	AGC target for dd-MS^2^	2e5	c-score Threshold	2.0
26.5	30	Maximum Injection time (ms) for dd-MS^2^	80	ID Quality filter	[A, B, C, D]
28.0	25	Precursor isolation window	1.2 *m*/*z*		
32.5	25	Dynamic exclusion (s)	2.5 (+), 3 (−)		

**Table 2 foods-12-02198-t002:** Potential lipid markers for discrimination of irradiated and non-irradiated Camembert.

Potential Lipid Markers	CAM_NI *	CAM_IRR *	Model
Negative			
Cer (d16:1_24:0)	21.60 ± 2.04	24.30 ± 3.05	PLS-DA
Cer (d18:1_23:0)	43.70 ± 2.42	46.20 ± 2.69	PLS-DA
Cer (d18:1_24:0)	41.60 ± 1.92	43.90 ± 1.89	PLS-DA
Hex1Cer (t35:2)	82.00 ± 34.00	51.10 ± 39.10	PLS-DA
LPC (18:2)	16.20 ± 17.50	3.54 ± 2.80	PLS-DA
LPE (18:2)	35.60 ± 22.10	12.70 ± 10.30	PLS-DA
PA (16:0_18:2)	190.00 ± 33.90	121.00 ± 87.00	PLS-DA
PA (18:2_18:2)	96.50 ± 19.30	62.30 ± 46.40	PLS-DA
PA (18:1_18:2)	130.00 ± 25.30	83.10 ± 56.20	PLS-DA
PC (18:3_18:3)	22.40 ± 17.20	3.15 ± 3.02	PLS-DA/LDA
PC (18:3_18:2)	74.40 ± 45.70	22.30 ± 18.30	PLS-DA/LDA
PC (18:2_18:2)	276.00 ± 139.00	119.00 ± 88.20	PLS-DA
PC (18:1_18:2)	127.00 ± 37.30	82.70 ± 29.30	PLS-DA
PE (16:1_18:2)	24.90 ± 10.80	11.80 ± 5.29	PLS-DA/LDA
PE (16:0_18:3)	38.30 ± 20.60	16.00 ± 4.55	PLS-DA/LDA
PE (16:1_18:1)	52.70 ± 14.80	42.30 ± 9.12	PLS-DA
PE (16:0_18:2)	273.00 ± 91.20	178.00 ± 50.00	PLS-DA
PE (18:3_18:3)	11.40 ± 8.65	1.49 ± 1.31	PLS-DA/LDA
PE (18:3_18:2)	43.00 ± 26.30	9.87 ± 7.16	PLS-DA/LDA
PE (18:2_18:2)	146.00 ± 70.30	50.50 ± 34.00	PLS-DA/LDA
PE (18:1_18:3)	45.20 ± 20.40	23.40 ± 2.80	PLS-DA
PE (18:1_18:2)	204.00 ± 36.30	165.00 ± 17.20	PLS-DA
PE (18:0_18:2)	145.00 ± 19.00	169.00 ± 26.00	PLS-DA
PI (15:0_18:2)	12.90 ± 6.14	5.77 ± 4.59	PLS-DA
PI (16:0_18:3)	14.70 ± 8.40	4.76 ± 3.78	PLS-DA/LDA
PI (16:0_18:2)	287.00 ± 121.00	133.00 ± 95.60	PLS-DA/LDA
PI (18:2_18:2)	23.50 ± 8.64	10.60 ± 8.08	PLS-DA/LDA
PI (18:1_18:2)	25.30 ± 8.74	16.70 ± 6.13	PLS-DA
PS (16:0_18:3)	16.00 ± 10.10	1.88 ± 0.98	PLS-DA/LDA
PS (16:0_18:2)	123.00 ± 55.80	49.80 ± 29.70	PLS-DA/LDA
Positive			
DG (8:0_14:0)	102.00 ± 23.10	73.30 ± 35.30	LDA
DG (10:0_14:0)	135.00 ± 26.10	89.10 ± 42.30	PLS-DA/LDA
DG (12:0_14:0)	151.00 ± 33.00	96.90 ± 52.00	PLS-DA/LDA
TG (11:2COOH_14:1_15:1)	279.00 ± 70.90	173.00 ± 50.30	PLS-DA/LDA
TG (11:3COOH_14:0_15:1)	278.00 ± 72.80	172.00 ± 48.50	PLS-DA/LDA
TG (11:3COOH_15:1_16:1)	200.00 ± 25.90	158.00 ± 30.90	PLS-DA/LDA
TG (18:1+O_18:1_18:1)	796.00 ± 158.00	953.00 ± 185.00	LDA
TG (18:2+O_18:0_18:0)	1080.00 ± 196.00	1280.00 ± 130.00	PLS-DA/LDA
TG (5:0CHO_12:0_12:0)	214.00 ± 57.20	129.00 ± 36.30	PLS-DA/LDA
TG (5:0CHO_12:0_14:1)	72.60 ± 23.10	40.30 ± 12.20	PLS-DA/LDA
TG (5:0COOH_12:0_12:0)	35.60 ± 4.06	24.40 ± 8.35	PLS-DA/LDA
TG (7:1CHO_12:0_12:0)	70.50 ± 22.40	39.60 ± 12.60	PLS-DA/LDA

(*) Mean of the sums of the peak areas (*n* = 12) (AU × 10^5^).

**Table 3 foods-12-02198-t003:** Diagnostic statistics for PLS-DA model.

	Double Cross-Validation	Bootstrap	Stratified Random Subsampling	Stratified Kennard–Stone Sampling	Kennard–Stone Sampling
Negative					
*RMSECV*	0.288	0.407	0.287	0.164	0.088
*Q^2^*	0.916	0.833	0.916	0.973	0.986
*DQ^2^*	0.940	0.926	0.944	1	0.991
Sensitivity	1	0.995	0.999	1	1
Specificity	0.999	0.989	0.996	1	1
Accuracy	0.998	0.992	0.998	1	1
Positive					
*RMSECV*	0.465	0.471	0.412	0.500	0.323
*Q^2^*	0.781	0.778	0.830	0.750	0.895
*DQ^2^*	0.845	0.861	0.890	0.751	0.909
Sensitivity	0.976	0.986	0.997	1	1
Specificity	0.982	0.990	1	1	1
Accuracy	0.988	0.988	0.999	1	1

## Data Availability

Data is contained within the article or supplementary material. Supplementary materials are available in a Mendeley Data repository doi:10.17632/4hf9n88733.1.

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
