# Peer review of "Untargeted Lipidomics and Chemometric Tools for the Characterization and Discrimination of Irradiated Camembert Cheese Analyzed by UHPLC-Q-Orbitrap-MS"

_foods, 2023, doi:10.3390/foods12112198_

Round 1

Reviewer 1 Report

This study is  a basic knowledge on lipid composition during post-processing of cheese, in this case under X-ray irradiation treatment basically focus on oxidised lipids.  However, there are some questions.

Comment:

1.      Why 3.0 kGy of X-ray radiation treatment was used in this study? Please state in your objective.

2.      Why you need to modify the extraction of fat from cheese samples using Folch method?

3.      In your opinion, why the results cannot be separated between these two samples using PCA alone?

4.       In the conclusion, you did mention that 3.0 kGy of X-ray radiation treatment is not enough to detect the oxidised lipid. In practical, is the X-ray radiation used can be more than 3.0 kGy?

5.      In your Introduction, you did mention “In this work, the lipid chemistry of commercial Camembert cheese”, I think you should change to lipid composition.

This study is  a basic knowledge on lipid composition during post-processing of cheese, in this case under X-ray irradiation treatment basically focus on oxidised lipids. The manuscript should be modify in certain parts.

Reviewer 2 Report

Line 29: remove italic text formatting from "or"

Line 31: delete [3], since it is repeated consecutively in line 34

Line 59: use L. monocytogenes instead Listeria monocytogenes, the abbreviated form can be used since its previously unabbreviated form was used

Line 27-63: the length of the paragraph is very large, commonly it is recommended that a paragraph should contain 6-17 lines

Line 119: add information (model, brand, and country) of the equipment used (vortex and centrifuge)

Line 119: use min instead of minutes

Line 120,121: use min instead of minutes

Line 125: 50 mg

Line 129: in subsubsections it is not necessary to use italic text formatting

Line 138: use 0–4 instead 0-4

Line 150: 70,000

Line 152: 17,500

Line 158,168: in subsubsections it is not necessary to use italic text formatting

Line 220: Results and Discussion?

Line 234: …5 phoshatidic...

Line 245,284,302,303,341,395, 401,410,417,437: in subsubsections it is not necessary to use italic text formatting

Line 246-282: the length of the paragraph is very large, commonly it is recommended that a paragraph should contain 6-17 lines

Line 332-335: check font size and font type

Line 399,430: 10,000

Line 435: 100%

Line 445: 30,000

Line 453: 6.6x10

Line 459: 1.0x10

References section

- the titles of each reference must be written in lowercase text format

- Line 505,508,570: scientific names must be written in italic text format

- Line 513: delete spaces …(2- DCB)…. (SPME)….(GC - MS).

Author Response

"Please see the attachment".

Reviewer 3 Report

The article "Untargeted Lipidomics and Chemometric Tools for the Characterization and Discrimination of Irradiated Camembert Cheese Analysed by UHPLC-Q-Orbitrap-MS". The manuscript is within the scope of the journal Foods. Overall, the manuscript is very well written, structured and informative. However, the hypothesis and goals to be achieved with the omic model are not clear. The final application of the extensive analytical work is not explicit in the article.

Please see my comments below.:

Abstract

Improve the wording of the objective and hypothesis of the paper.

Improve the formulation of the application of your results. How your results could be used directly in the control of irradiated foods. What lipids did you find as markers?

Introduction

A large part of the introduction is devoted to food safety and foodborne pathogens. However, the article is not directly related to these pathogens. Again, the introduction does not make clear how your results are to be applied.

Material and Methods

The number of initial cheeses and then portions performed is low enough to have reliable lipidomics results. Please explain why you did not use a larger number of samples in your model.

Results and discussion

It would be useful to include a section and table on lipid identification. At least of the 42 selected as potential markers. This table should include the mass of the compounds, error, molecular formula and their fragmentation pattern.

Author Response

"Please see the attachment".
